# Mitochondrial Metabolism in X-Irradiated Cells Undergoing Irreversible Cell-Cycle Arrest

**DOI:** 10.3390/ijms24031833

**Published:** 2023-01-17

**Authors:** Eri Hirose, Miho Noguchi, Tomokazu Ihara, Akinari Yokoya

**Affiliations:** 1College of Science, Ibaraki University, 2-1-1 Bunkyo, Mito 310-8512, Japan; 2Institute for Quantum Life Science, National Institutes for Quantum Science and Technology, 4-9-1 Anagawa, Chiba 263-8555, Japan; 3Graduate School of Science and Engineering, Ibaraki University, 2-1-1 Bunkyo, Mito 310-8512, Japan

**Keywords:** cell-cycle arrest, mitochondria, membrane potential, radiation effect, X-rays

## Abstract

Irreversible cell-cycle-arrested cells not undergoing cell divisions have been thought to be metabolically less active because of the unnecessary consumption of energy for cell division. On the other hand, they might be actively involved in the tissue microenvironment through an inflammatory response. In this study, we examined the mitochondria-dependent metabolism in human cells irreversibly arrested in response to ionizing radiation to confirm this possibility. Human primary WI-38 fibroblast cells and the BJ-5ta fibroblast-like cell line were exposed to 20 Gy X-rays and cultured for up to 9 days after irradiation. The mitochondrial morphology and membrane potential were evaluated in the cells using the mitochondrial-specific fluorescent reagents MitoTracker Green (MTG) and 5,5′,6,6′-tetraethyl-benzimidazolylcarbocyanine iodide (JC-1), respectively. The ratio of the mean MTG-stained total mitochondrial area per unit cell area decreased for up to 9 days after X-irradiation. The fraction of the high mitochondrial membrane potential area visualized by JC-1 staining reached its minimum 2 days after irradiation and then increased (particularly, WI-38 cells increased 1.8-fold the value of the control). Their chronological changes indicate that the mitochondrial volume in the irreversible cell-cycle-arrested cells showed significant increase concurrently with cellular volume expansion, indicating that the mitochondria-dependent energy metabolism was still active. These results indicate that the energy metabolism in X-ray-induced senescent-like cells is active compared to nonirradiated normal cells, even though they do not undergo cell divisions.

## 1. Introduction

One of the major remedies for cancer treatment is radiation therapy, and it has been shown that radiation exposure causes irreversible G1-arrest or premature senescence in normal human cells [1,2,3]. Furthermore, several studies have demonstrated therapy-induced senescence [4,5] in which normal stromal cells as well as cancer cells were irreversibly arrested in the G1 phase and manifested senescence phenotypes [6,7,8]. Since senescent stromal cells and cancer cells might trigger change in the tissue microenvironment through an inflammatory response of the cells in the exposed organs, they could potentially be involved in the development of secondary carcinogenesis. Therefore, from a cancer management point of view, it is critical to understand the biology underlying irreversible cell-cycle arrest. X-ray stress, as a typical ionizing irradiation triggering the expression of p21, which functions downstream of the p53 pathway, inhibits the pathway of the cyclin E-cyclin-dependent kinase 2 (CDK2) complex [9]. p16 inhibits the cyclin D-CDK4/6 complex [10]. Inhibition of these cyclin-CDK pathways leads to irreversible cell-cycle arrest, because these cyclin-CDK complexes activate the transcription factor E2F, which is required for cell-cycle progression [11]. More recently, not simple but rather complex processes have been considered as factors regulating irreversible cell-cycle arrest [4,5,12,13].

Mitochondria also play crucial roles in irreversible cell-cycle arrest through the increased production of X-ray–induced reactive oxygen species (ROS) [14]. ROS can decrease mitochondrial function and induce irreversible cell-cycle arrest [15] by the following process: the mitochondrial permeability transition pore opens, the NAD+/NADH ratio decreases [16], AMP-activated protein kinase is activated, p21 expression is upregulated by AMP-activated protein kinase-induced p53 [17], and the expression of other downstream genes of CDK inhibitors is induced [18]. Therefore, for the comprehensive understanding of irreversible cell-cycle arrest biology, it is essential to clarify mitochondrial energy metabolism during the progression of irreversible cell-cycle arrest. Previously, it was reported that the total mitochondrial area was increased in hypertrophic cells (i.e., irreversible cell-cycle-arrested cells). Hypertrophy occurs when cells avoid entry into mitosis and form tetraploids [19], and is due to the increase in mitochondria as the cell size expands. ATP production is also expected to increase in hypertrophic senescent cells, since eukaryotic cells attempt to maintain their optimal size to maintain homeostasis, and cellular contents, such as mitochondria, are scaled linearly with cell size [20]. Nevertheless, irreversible cell-cycle-arrested cells require less ATP compared with normal cells, because 60% of the total energy production from aerobic metabolism is used per cell cycle [21], which should be validated in radiation-induced irreversible cell-cycle arrest.

In the present study, we characterized X-irradiation-induced irreversible cell-cycle-arrested cells through examining the mitochondrial content using MitoTracker Green (MTG) staining and the mitochondrial membrane potential (ΔΨm) using 5,5′,6,6′-tetraethyl-benzimidazolylcarbocyanine iodide (JC-1) staining. Because ionizing radiation prematurely induces irreversible cell-cycle arrest in stromal cells, namely, fibroblasts [22,23], we used WI-38 cells exposed to 20 Gy X-rays to examine the effect of irradiation on fibroblasts. We also investigated immortalized BJ-5ta cells, which do not show endogenous senescence caused by telomere shortening, to study irreversible cell-cycle arrest caused by radiation alone. Since one of the characteristics of refractory cancers, such as pancreatic cancer and glioblastoma, is the presence of a substantial number of stromal cells [24,25,26,27], understanding the radiation response of stromal fibroblasts is essential for radiation therapy for better cancer management.

## 2. Results

### 2.1. Cell Proliferation against X-ray Dose

The manifestation of irreversible cell-cycle arrest was examined in WI-38 cells by counting the number of nonirradiated cells (0 Gy) and 20 Gy X-irradiated cells (Figure 1). Nonirradiated WI-38 cells showed a 10-fold increase in proliferation for up to 9 days of incubation, while cell proliferation was significantly suppressed in 20 Gy X-irradiated WI-38 cells. Similarly, nonirradiated BJ-5ta cells also showed a significant increase in proliferation for up to 7 days of incubation and then reached confluence, but the proliferation of 20 Gy X-irradiated cells was significantly lower than that of nonirradiated cells.

For certain tumor cell lines, exposure to 20 Gy induces the significant elimination of cells [28]. However, WI-38 and BJ-5ta cells did not show such a reduction in cell number even after 20 Gy, and most of the cells (90%) continued to exist in culture without cell-cycle progression. The result was in agreement with our previous study [29], confirming that 20 Gy efficiently induced cell cycle arrest without cell loss. Therefore, we determined that 20 Gy was an appropriate dose to induce irreversible cell-cycle arrest.

### 2.2. Validation of Irreversible Cell-Cycle Arrest by EdU and SA-β-Gal Staining

Irreversible cell-cycle arrest in WI-38 cells after exposure to 20 Gy X-irradiation was confirmed by EdU-labeling experiments. In addition, senescence was verified by SA-β-gal staining, which was used to validate the senescence induction [30]. As shown in Figure 2A, the majority of cells after irradiation were negative for EdU staining. The percentage of EdU-positive cells was plotted against time after irradiation (Figure 2B), and all values after irradiation were < 2%. Meanwhile, a blue color stain for SA-β-gal developed in a time-dependent manner after irradiation (Figure 3A). The percentage of SA-β-gal-positive cells in three randomly chosen microscopic fields increased in a time-dependent manner to approximately 90% of cells at 9 days after irradiation (Figure 3B). The BJ-5ta cells also showed a similar tendency as WI-38 cells. From these findings, we confirmed that 20 Gy X-irradiation induced irreversible cell-cycle arrest, which has often been referred to as premature senescence.

### 2.3. Mean Mitochondrial Area after Irradiation Determined by MTG Staining

The analysis of MTG-stained mitochondria showed that the total mitochondrial area increased concurrently with the continuous hypertrophic changes in the cell shape after irradiation (Figure 4A). The mean total mitochondrial area per cell (Figure 4B) and mean cell area determined in phase contrast images (Figure 4C) were plotted against incubation time after irradiation. For WI-38 and BJ-5ta cells, there was a continuous increase in the total mitochondrial area and cell area over time. While the ratio of mitochondria area against cellular area, which was calculated by dividing the mean total mitochondrial area by cell area (Table 1), decreased after irradiation in WI-38, the ratio of BJ-5ta cells did not show a significant decrease.

### 2.4. High ΔΨm Area Determined by JC-1 Staining

The low ΔΨm area (green) of each X-irradiated cell line increased with the hypertrophic changes in the cell shape. Meanwhile, the high ΔΨm area (red) did not increase in a time-dependent manner after irradiation (Figure 5A). The fraction of the high ΔΨm area in the total JC-1-stained area normalized by that of control cells was plotted against time after irradiation (Figure 5B). The minimum fraction value of the WI-38 cells was observed at 2 days after irradiation and then increased sharply (1.8-fold of the value of the control). In contrast, the BJ-5ta cells showed a slight minimum at 2 days after irradiation and increased slowly thereafter but did not surpass the control level.

## 3. Discussion

The irreversible cell-cycle-arrested cells used in this study shared many characteristics with senescent cells [31], such as the permanent loss of proliferative potential determined by EdU-staining (Figure 2), an enlarged cell size and flattened morphology (Figure 3), and the expression of SA-β-gal (Figure 3), confirming that both WI-38 and BJ-5ta induce senescence through irreversible cell-cycle arrest. Thus, the WI-38 and BJ-5ta cells, two representative cell lines of fibroblasts used in the present study, were able to induce premature senescence and could sustain their metabolism without any apoptotic features even after high-dose X-ray exposure (20 Gy) (Figure 1). Although it remains to be clarified with various normal cells with different tissue origin whether irreversible cell-cycle arrest is a general feature of X-irradiated normal cells, we confirmed that WI-38 and BJ-5ta cells entered a state of irreversible cell-cycle arrest.

This study is the first to report the mitochondrial metabolism changes in irreversible cell-cycle-arrested cells. Many studies have reported the effects of acute radiation on mitochondria in terms of dysfunction, as revealed by fragmentation, ROS generation, ATP concentration and mitochondrial gene expression (for a review, see [32] and references therein). However, little is known about the mitochondria in the irreversible cell-cycle-arrested cells. Our finding of an increase in the MTG-stained total mitochondrial area in the WI-38 and BJ-5ta cells for up to 9 days after irradiation suggests that the production of mitochondria exceeded the removal of damaged mitochondria (Figure 4B), presumably through the autophagy process. In addition to the increase in the mitochondrial area, the volume of both cells also showed an increase, presumably because of the reproduction of cellular components without undergoing cell divisions (hypertrophic effect). While the ratio of dividing the total mitochondrial area by the cell area (Table 1) for both cells decreased continuously, and WI-38 cells in particular reached approximately half of the control at 9 days after irradiation, it was suggested that the reproducing mitochondria were less vibrant than those for the other cellular components.

The normalized fraction of the high ΔΨm area of the WI-38 and BJ-5ta cells visualized by JC-1 staining reached its lowest level 2 days after irradiation (Figure 5B). This observation suggests that the mitochondrial activity was temporarily decreased due to the degradation of damaged mitochondria (as indicated above), as reported previously [14], and then the high ΔΨm area was re-established within a few days. Although the high ΔΨm area was overshot for the primary cells (WI-38) 5 days after irradiation, the fraction of the high ΔΨm area was retained for the immortalized BJ-5ta cells. A low fraction of the high ΔΨm area was also reported for other immortalized human fibroblast cells, BJ-1 h-TERT [33]. This evidence strongly suggests that the energy metabolism is upregulated in a delayed manner in primary cells, and this is not the case for immortalized cells. Conversely, the manipulation of immortalization might alter the energy metabolism relevant to stress responses.

Because the mitochondrial genome encodes less than 1% of mitochondrial proteins [34], nuclear DNA is responsible for producing the proteins or chaperones required for mitochondria synthesis. To our knowledge, no precise measurements have been made to determine the time required for the complete recovery of mitochondrial assembly. However, because a complex double membrane structure and various membrane protein complexes need to be assembled, we estimate that it would take at least one cell cycle, normally 16–24 h. Accordingly, it is reasonable to assume that the cells temporarily decreased their normalized fraction of the high ΔΨm area and then prioritized the expansion of existing mitochondrial activation sites to address their ATP requirements.

Interestingly, in WI-38 cells, the normalized fraction of the high ΔΨm area increased markedly (1.8-fold of the value of the control) at 5 and 9 days after irradiation (Figure 5B). Because ΔΨm is formed by the electrochemical proton gradient for ATP synthesis [35,36], this noticeable increase in the fraction of the high ΔΨm area is possibly a response to increased ATP demand at 5 days after irradiation, because it has now become clear that irreversible cell-cycle-arrested cells require more bioenergy [37]. The requirement for extra ATP could be caused by the overexpression of β-gal, senescent cell-specific heterochromatin structures (senescence-associated heterochromatic foci), activation of the p53-p21 and p16-Rb pathways to arrest the cell cycle, or avoidance of mitosis and transition to the G1 phase. Furthermore, because irreversible cell-cycle-arrested cells share many characteristics with senescent cells, among which the synthesis of senescence-associated secretory phenotype (SASP)-related proteins could be the most relevant, they really need ATPs. A previous study reported that 41% or 33% of genes in proliferating cells were up- or downregulated after irreversible cell-cycle arrest, and their expression was reversed to be down- or upregulated due to the fact of mitochondrial depletion, respectively [38]. They found that the regulated genes included many SASP-related genes. In terms of gene regulation, this indicates that mitochondria contribute to the homeostasis of irreversible cell-cycle-arrested cells through the production of extra ATP for the SASP.

The chronological changes in the relative mitochondrial area per cell volume and high ΔΨm area for immortalized BJ-5ta cells, in which their telomere was manipulated to ensure their unrestricted cell divisions, were not significant when compared with those for WI-38 cells. A link between telomere function and mitochondria may underlie those different responses between the cell lines. Sahin et al. [39] revealed the presence of mitochondrial dysfunction in mice that were engineered to progressively lose telomere function. This was the result of the reduced activity of PGC-1α and PGC-1β, which are key factors regulating mitochondrial function. Reducing the activity of these factors leads to a decrease in the mitochondrial number and metabolic failure of irreversible cell-cycle-arrested cells. In other words, the mitochondria in BJ-5ta cells might be less sensitive to X-ray irradiation, which should be determined in future experiments.

To reveal the precise correlation between the mitochondrial effects and the maintenance of cell function in irreversible cell-cycle-arrested cells, further studies are also needed. In particular, inhibiting ΔΨm with uncoupling agents such as carbonyl cyanide m-chlorophenyl hydrazone and using normal human lung fibroblast cells (MRC5 and TIG-3), which have been used extensively in previous studies, or *ρ0* cells [40], which do not have mitochondrial DNA, would be useful approaches. We might need to immortalize WI-38 cells to reveal more about the connections between mitochondria and telomere function. Furthermore, the quantification of ATP production in cells as an energy metabolism index, oxygen consumption rate and extracellular acidification rate in live cells as indices for mitochondrial respiration and glycolysis, will clearly show the effect of X-irradiation on the respiratory chain function.

In summary, we have described the mitochondrial morphology in irreversible cell-cycle-arrested cells exposed to high-dose X-rays. Their chronological changes indicate that the mitochondrial volume in the senescent-like cells showed an increase, indicating that they became highly active to meet the increase in the ATP requirements of the cells. Contrary to what was previously thought, our findings suggest that the cells are not less vigorous but intensely active, thus they may impact upon surrounding normal cells through their high energy metabolic products, such as SASP. Our study serves as a window to an understanding of the role of long-lived cell-cycle-arrested cells, namely, senescent cells, in organs or the whole body. Since premature senescence in the tumor microenvironment has been extensively discussed in terms of the treatment efficacy and manifestation of adverse effects, as well as the QOL of the patients [41,42,43,44,45,46], a comprehensive understanding of the radiation biological significance of irreversible cell-cycle arrest after radiotherapy is expected.

## 4. Materials and Methods

### 4.1. Cell Lines and Culture Conditions

Human primary fetal lung fibroblast cells (WI-38; American Type Culture Collection, Manassas, VA, USA) and human hTERT-immortalized foreskin fibroblast-like cells (BJ-5ta; American Type Culture Collection) were used. WI-38 cells were grown in Dulbecco’s modified Eagle’s medium with l-glutaminase and phenol red (D-MEM; Fujifilm, Tokyo, Japan) supplemented with 10% fetal bovine serum (Thermo Fisher Scientific, Waltham, MA, USA) and 1% antibiotic–antimycotic (Thermo Fisher Scientific, Waltham, MA, USA). The BJ-5ta cells were grown in a 4:1 mixture of D-MEM and Medium 199 (Sigma-Aldrich, St. Louis, MO, USA), supplemented with 10% fetal bovine serum (Thermo Fisher Scientific, Waltham, MA, USA) and 1% antibiotic–antimycotic (Thermo Fisher Scientific, Waltham, MA, USA). The cells were cultured at 37 °C in a humidified incubator with an atmosphere of 95% air and 5% CO_2_.

The experiments were performed three times independently using cells seeded 2 days before irradiation. Approximately 2.0 × 10^5^ cells/well were seeded on Falcon T25 flasks (Corning, Inc., Corning, NY, USA) for the measurement of cell proliferation; approximately 2.0 × 10^4^ cells/well were seeded on Falcon plastic 96-well plates (Corning, Inc., Corning, NY, USA) for 5-ethynyl-2′-deoxyuridine (EdU) staining. Approximately 5.0 × 10^4^ cells/dish were seeded on Falcon plastic 35 mm cell culture dishes (Corning, Inc., Corning, NY, USA) for senescence-associated β-galactosidase (SA-β-gal) staining, and approximately 5.0 × 10^4^ cells/dish were seeded on Iwaki glass 35 mm cell culture dishes (Iwaki Co., Fukushima, Japan) for MTG and JC-1 staining. The growth medium was replaced with fresh medium before irradiation, and the cells were incubated without changing the medium.

### 4.2. X-Irradiation

The cells were exposed to X-rays using an X-ray generator with a tungsten target (M-150WE; Softex, Ebina, Japan). The energy of the characteristic X-rays was 60–70 keV. An aluminum filter with a 0.2 mm thickness was used to minimize the low-energy X-rays (<1.5 keV). The X-ray generator was operated at a tube voltage of 150 kVp and a tube current of 3.48 mA, corresponding to a dose rate of 1.0 Gy/min. The cells were irradiated for 20 min to provide a dose of 20 Gy.

### 4.3. Observation of Living Cells Using a Fluorescence Microscope

The cells were observed under a BZ-710X fluorescence microscope (Keyence, Osaka, Japan). The excitation and emission wavelengths were 360 and 460 nm for 4′,6-diamidino-2-phenylindole (DAPI), 490 and 516 nm for MTG, and 485 and 535 nm (green fluorescence) or 520–570 and 570–610 nm (red fluorescence) for JC-1, respectively. The absorption filters (Keyence) for DAPI (460 ± 50 nm), green fluorescent protein (525 ± 50 nm), and tetramethylrhodamine-5-(and -6-) isothiocyanate (605 ± 70 nm) were used for green, red and blue fluorescence, respectively. Phase contrast images were taken along with the fluorescence observations. During the observations, the cells were cultured in a humidified chamber (Tokai Hit, Shizuoka, Japan) set in the fluorescence microscope at 37 °C and 5% CO_2_. A lens of 20× magnification for EdU, 10× magnification for SA-β-gal, or an oil-immersion objective lens of 60 × magnification for MTG and JC-1 was set approximately 30 min before the observations to keep the temperature of the lens constant. Observations were performed with a BZ-X Viewer microscope control application software (Keyence).

### 4.4. Measurement of Cell Proliferation

Nonirradiated and 20 Gy X-irradiated WI-38 and BJ-5ta cells were cultured in an incubator for up to 9 and 15 days, respectively. The cells were washed with phosphate-buffered saline (PBS) (-) (Wako, Tokyo, Japan), treated with 0.05% trypsin-EDTA (Themo Fisher Scientific, Waltham, MA, USA) for 3 min in a cell incubator, and resuspended in serum-containing complete medium to deactivate the trypsin. The cells were counted with a Coulter counter (Beckman Coulter, Brea, CA, USA). The BJ-5ta cells were cultured longer than WI-38 cells, because they are an hTERT immortalized cell line, and this might result in some molecular differences compared to primary cells (i.e., WI-38).

### 4.5. EdU Staining

EdU is a thymidine analog that is an alternative to 5-bromodeoxyuridine. An EdU Proliferation Kit (Abcam, Cambridge, UK) was used to detect and quantify the cell proliferation. The cells were cultured in an incubator with an EdU solution for 2 h. A fixative solution was added to each well, and the cells were incubated again for 15 min at room temperature. The culture plate was protected from light exposure during incubation. The cells were washed twice with a wash buffer and stored in PBS in an incubator to prevent the cells from drying out. After the cells in all conditions were fixed, they were stained on the same day, as follows: A permeabilization buffer was added to the cells, and they were incubated for 20 min at room temperature. After the cells were washed twice with the wash buffer, a reaction mix was added to stain the cells, and they were incubated for 30 min at room temperature. The cells were washed again with the wash buffer and PBS, and the cell nuclei were labeled with DAPI (Dojindo, Kumamoto, Japan). Fresh PBS was added for the EdU-positive cell observations under a microscope.

### 4.6. SA-β-Gal Staining

Irreversible cell-cycle-arrested cells can be identified by SA-β-gal staining [47]. A senescence cells histochemical staining kit (Sigma-Aldrich, St. Louis, MO, USA) was used for SA-β-gal staining. After the cells were washed twice with PBS, they were fixed with a fixation buffer and incubated for 7 min at room temperature. The cells were washed three times with PBS and stored at 4 °C in PBS with the dishes covered with parafilm to prevent the cells from drying out. After the cells in all conditions were fixed, a staining mixture was added, and they were stained on the same day. The dishes were covered with parafilm during cell staining, preferably in the absence of CO_2_, as this can lower the pH of the buffer and affect staining. The cells were incubated overnight in a cell incubator, washed three times with PBS, and fresh PBS was added to the dishes. Then, the cells were observed under a microscope. The number of SA-β-gal-positive (blue) cells versus the total number of cells was scored to calculate the percentage of SA-β-gal-positive cells. The number of cells in a field of view was counted per condition.

### 4.7. MTG Staining

The MTG chemical probe (Invitrogen, Carlsbad, CA) was used for mitochondrial area analysis. MTG labels mitochondria with green fluorescence regardless of ΔΨm. MTG was diluted with DMSO to adjust the concentration to 100 µM as a stock solution and stored at −20 °C prior to use. The frozen stock solution was equilibrated at room temperature and mixed with D-MEM to a final concentration of 200 nM as a staining solution. D-MEM was removed from the glass dish on which the cells were cultured, and the staining solution was added. After 20 min of incubation in a cell incubator, the solution was removed from the dish and the cells were washed twice with PBS. Fresh D-MEM was applied for observations under a fluorescence microscope.

### 4.8. JC-1 Staining

The JC-1 chemical probe was used to visualize ΔΨm. A commercial JC-1 kit (Cayman Chemical, Ann Arbor, MI) was used that comprises a monomer that yields green fluorescence at low ΔΨm (≤−100 mV) and a polymer that yields red fluorescence at high ΔΨm (≥−140 mV). The frozen JC-1 reagent was equilibrated to room temperature. The dispensed reagent was mixed with 225 µL D-MEM as a stock solution and stored at −20 °C prior to use. After thawing 25 µL of the stock solution, it was mixed with 500 µL D-MEM as a working solution. The D-MEM was removed from the glass culture dish, and the working solution was added to the cells in a cell incubator for 20 min. After incubation, the solution was removed from the dish and the cells were washed twice with PBS. Fresh D-MEM was applied for observations under a fluorescence microscope.

## Figures and Tables

**Figure 1 ijms-24-01833-f001:**
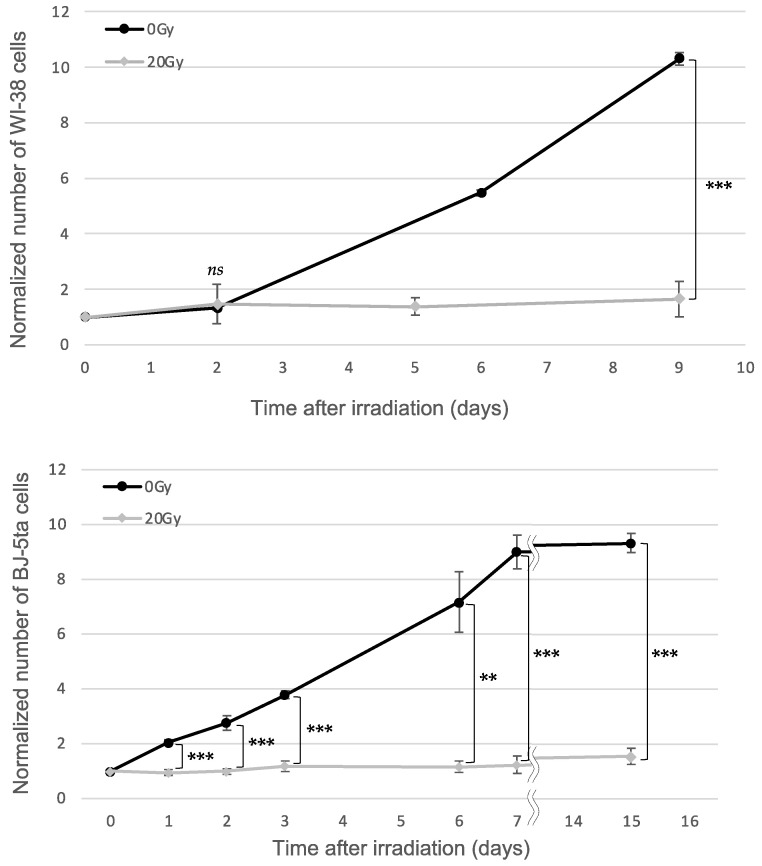
Cell proliferation against the X-ray dose. Nonirradiated and 20 Gy X-irradiated cells (seeded at 2.0 × 10^5^ cells/well) measured over a 9-day period for WI-38 cells and a 15-day period for BJ-5ta cells. All values were normalized by the number of cells at day 0, and the error bars represent the standard deviation (*n* = 3). The symbol * indicates the *p*-values of the Student’s t-test compared with the control and irradiated samples, respectively; *** *p*  <  0.001, ** *p* < 0.01, and *ns* was not significant.

**Figure 2 ijms-24-01833-f002:**
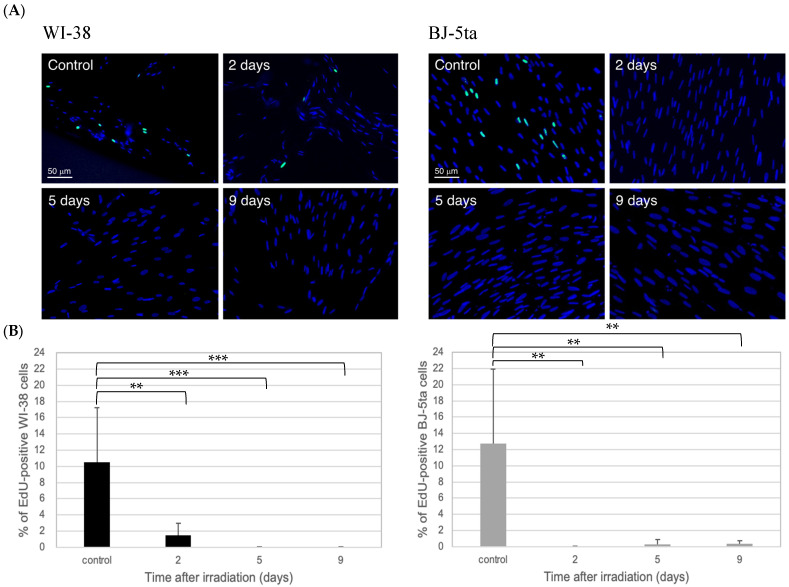
(**A**) Representative microscopic images of EdU–and DAPI-stained cells. Nonirradiated (control) and 20 Gy X-irradiated WI-38 and BJ-5ta cells observed under a fluorescence microscope at 20 × magnification. The images of the irradiated cells were taken at 2, 5, and 9 days after irradiation. EdU was added for 2 h before analysis. A significant decrease in the percentage of EdU-positive green cells indicates that irreversible cell-cycle arrest has occurred. The cells were counterstained with DAPI (blue). The scale bar was common for the images of each cell. (**B**) Quantification of cell proliferation based on EdU uptake in nonirradiated (control) and irradiated WI-38 and BJ-5ta cells. The number of EdU-positive WI-38 and BJ-5ta cells divided by the total number of cells in three microscopic fields are plotted against the incubation time. Approximately 200 cells were scored in one microscopic field. The error bars represent the standard deviation. Statistical analysis was performed with one-way ANOVA (*p*  <  0.001) plus a Student’s t-test. The symbol * indicates the *p*-values of the *t*-test compared with the control and irradiated samples, respectively; *** *p*  <  0.001 and ** *p* < 0.01.

**Figure 3 ijms-24-01833-f003:**
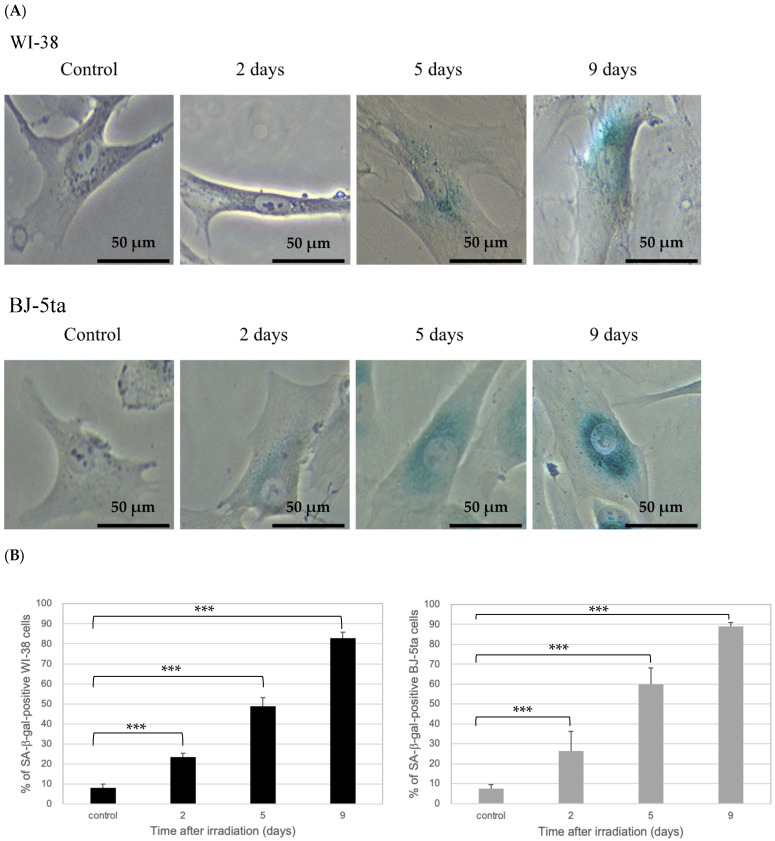
(**A**) Representative microscopic images of SA-β-gal-stained nonirradiated (control) and 20 Gy X-irradiated WI-38 and BJ-5ta cells observed under a fluorescence microscope at 20 × magnification. The images of the irradiated cells were taken at 2, 5, and 9 days after irradiation. Blue staining indicates irreversible cell-cycle-arrested cells. (**B**) The fraction of the SA-β-gal-positive WI-38 and BJ-5ta cells normalized by the total number of cells at each incubation time. Approximately 70 cells were scored in one microscopic field. The error bars represent the standard deviation. Statistical analysis was performed with one-way ANOVA (*p*  <  0.001) plus a Student’s t-test. The symbol * indicates the *p*-values of the t-test compared with the control and irradiated samples, respectively; *** *p*  <  0.001.

**Figure 4 ijms-24-01833-f004:**
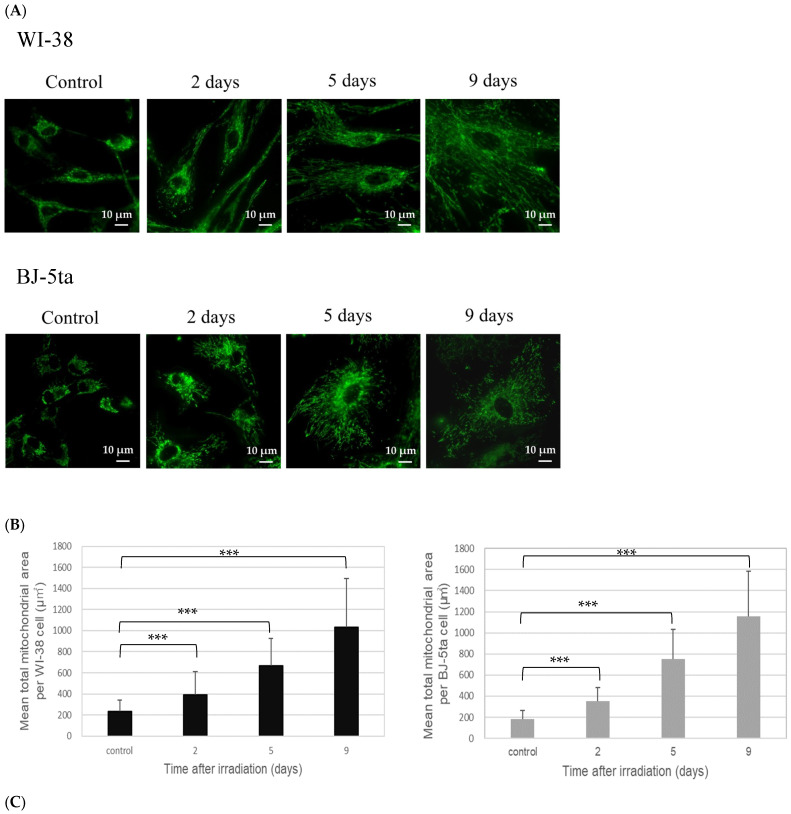
(**A**) Representative images of MTG-stained mitochondria of nonirradiated (control) and 20 Gy X-irradiated WI-38 and BJ-5ta cells observed under a fluorescence microscope at 60 × magnification. The images of the irradiated cells were taken at 2, 5, and 9 days after irradiation. (**B**) Mean total mitochondrial area per cell. (**C**) Mean cell area (from phase contrast images). The error bars in (**B**) and (**C**) represent the standard deviation (*n* = 90). Statistical analysis was performed with one-way ANOVA (*p*  <  0.001) plus a Student’s t-test. The symbol * indicates the *p*-values of the *t*-test compared with the control and irradiated samples, respectively; *** *p*  <  0.001.

**Figure 5 ijms-24-01833-f005:**
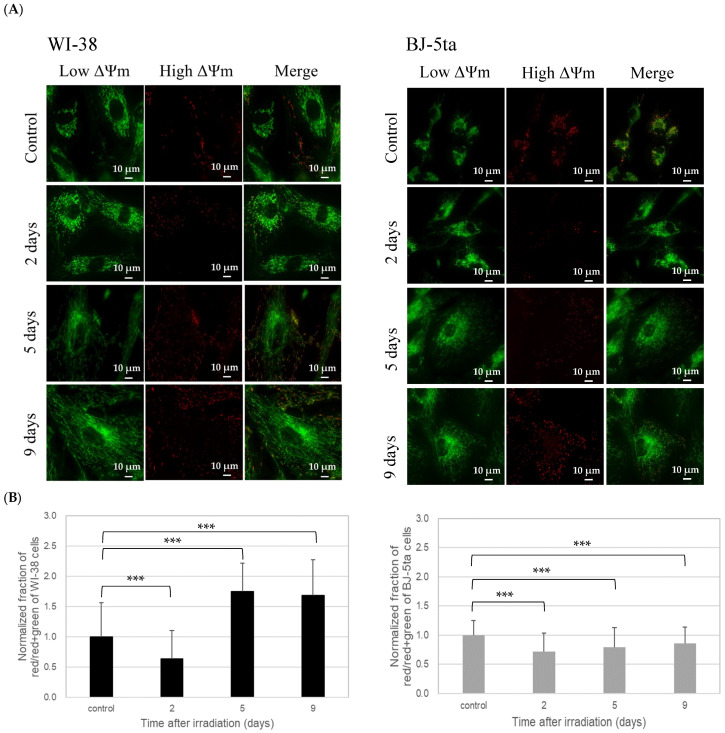
(**A**) Representative images of the low and high ΔΨm regions with JC-1 staining of nonirradiated (control) and 20 Gy X-irradiated WI-38 and BJ-5ta cells observed under a fluorescence microscope at 60× magnification. The images of the irradiated cells were taken at 2, 5, and 9 days after irradiation. (**B**) Fraction of the high ΔΨm area defined as the ratio of the red area to the sum of the red and green areas. All values were normalized by the control cells, and the error bars represent the standard deviation (*n* = 90). Statistical analysis was performed with one-way ANOVA (*p*  <  0.001) plus a Student’s t-test. The symbol * indicates the *p*-values of the *t*-test compared with the control and irradiated samples, respectively; *** *p*  <  0.001.

**Table 1 ijms-24-01833-t001:** Averaged geometric data of nonirradiated (control) and 20 Gy X-irradiated WI-38 and BJ-5ta cells. The total mitochondrial area was obtained by MTG staining.

Time After Irradiation (Days)	Control	2	5	9
WI-38				
Total mitochondrial area (μm^2^)	237 (±104)	390 (±221)	668 (±258)	1034 (±459)
Cell area (μm^2^)	2290 (±1220)	5270 (±2810)	9770 (±4470)	21,100 (±11,700)
Ratio (Total mitochondrial area/cell area)	0.10	0.074	0.068	0.049
BJ-5ta				
Total mitochondrial area (μm^2^)	181 (±86)	351 (±131)	752 (±283)	1155 (±431)
Cell area (μm^2^)	2280 (±809)	4870 (±1790)	11,200 (±5320)	17,400 (±7220)
Ratio (Total mitochondrial area/cell area)	0.079	0.072	0.067	0.067

The cell area was determined from the phase contrast images. The ratio represents the mean total mitochondrial area per unit cell area. The errors of plus or minus represent the standard deviation (*n* = 90).

## Data Availability

The data presented in this study are available upon request from the corresponding author.

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
