# Peer review of "Mitochondrial Metabolism in X-Irradiated Cells Undergoing Irreversible Cell-Cycle Arrest"

_ijms, 2023, doi:10.3390/ijms24031833_

Round 1
Reviewer 1 Report (Previous Reviewer 1)
I am OK with this new version and the changes made.Author Response
To reviewer #1,
Thank you very much for your positive evaluation to our manuscript. We are asking the editor to transfer the manuscript to English Editing Service of the journal.
Reviewer 2 Report (New Reviewer)
Dear authors,
I read your work with great pleasure. I am sure it will be a milestone for future research on all those radiobiological effects of radiation that we do not yet know and which will have critical clinical implications, both by improving therapeutic efficacy and by reducing the toxicity of treatments.
Best wishes for your scientific research.
Dr. D. Anzellini
Author Response
Dear Dr. D. Anzellini, as the reviewer #2,
Thank you very much for your positive evaluation to our revised manuscript. We would like to forward our project focusing much on molecular mechanism in the future.
Reviewer 3 Report (New Reviewer)
The article "Mitochondrial Metabolism in X-irradiated Cells Undergoing Irreversible Cell-Cycle Arrest" by Eri Hirose et al. The article deals with an interesting and still open topic: What happens when cells undergo a senescence process that, regardless of the cause behind it, has irreversible cell-cycle arrest as its main effect?
The authors report in the abstract, "Cells undergoing irreversible cell cycle arrest that do not undergo cell division are believed to be significantly inactive in terms of energy metabolism due to unnecessary energy consumption for cell division.". They add that evidence has been obtained in their study that overturns this prevailing orthodoxy. Their statement is too simplistic and is not reflected in the more recent but established literature. Many papers in the literature report the complex process that accompanies cell cycle arrest induced by radiation exposure and in general when senescence processes occur.
Many articles and reviews report the different mechanistic pathways and mediators underlying the finite proliferation of normal somatic cells and how entry into senescence leading to stable cell cycle arrest is regulated.
The authors are urged to downplay this claim, which is not reflected in the current literature.
Thus, the paper does not address a new and scientifically relevant topic and that said, the work loses much of the claimed scientific originality, but it may be still an interesting work because it proposes a possible marker, namely the different dependance of Delta membrane potential DΨ of dose exposure. This part of the study is certainly more interesting and deserves more in-depth study than the observational-only description.
The authors should stress this part and further investigate the possible implications of DΨ behavior as a function of time after irradiation and better justify the different behavior of the two lines that had instead shown close similarities for mean mitochondrial area and validation tests of irreversible cell-cycle arrest by EdU and SA-β-gal staining, after irradiation
Furthermore, although the circumvention of senescence and the acquisition of unlimited replicative potential is a key event necessary for malignant transformation, the underlying signaling pathways and the basis for the stability of growth arrest are poorly understood. Therefore, this paperi is welcome. Greater understanding is therefore essential if we are to prevent tissue dysfunction without increasing the risk of developing cancer.
The authors are invited to make a connection with the possible implications of their findings in terms of preventing carcinogenesis.
There is also much room for further progress in understanding the mechanisms underlying transient and short-lived senescence that promotes tissue development, regeneration, and repair, as it is less well characterized than the deleterious effects of stable senescence. One of the major obstacles in the field of senescence is the lack of a single universal and robust biomarker that can identify senescent cells with high sensitivity and specificity and that can differentiate them from terminally differentiated, quiescent, and other nondividing cells. Growth arrest is a key feature that can be easily demonstrated in vitro using colony formation assays or BrdU/EdU incorporation assays that measure DNA synthesis. However, measurement of DNA synthesis is not entirely specific, as DNA repair may still be active. Measuring the expression levels of CDKIs p16INK4A and p21WAF1=CIP1 is critical for detecting cell cycle arrest but is not persistently expressed by senescent cells. The accumulation of high levels of p16INK4A is necessary to maintain the senescent state by preventing RB inactivation, which allows it to be used extensively as a senescence marker in most normal nontransformed cells and tissues. Because of the heterogeneous and dynamic nature of senescence, there is currently no single totally reliable biomarker. Recently, a three-step multimarker workflow has been proposed to accurately detect senescent cells.
The first stage includes the assessment of senescence activity associated with beta-galactosidase (SA-b-gal) and/or lipofuscin (GL-13 or SBB) accumulation. The authors reported experiments on the detection of SA-beta-gal).
The second phase examines frequently observed markers of senescent cells, including transcriptional signatures related to cell cycle arrest and SASP, such as increased expression of cyclin-dependent kinase inhibitors and a subset of SASP genes, along with decreased expression of proliferation markers such as cyclins, CCNA2 and CCNE2, and LMNB1. A deeper understanding of the underlying mechanisms that regulate senescence will provide promising translational opportunities to develop novel therapeutic approaches that minimize the harmful consequences of senescence. Targeting senescence using senolytics to selectively eliminate senescent cells or modulating SASP using small molecules or antibodies will not only help treat senescence-related diseases, but may help improve the healthspan of individuals.
A thorough understanding of the regulation of SASP is needed to harness it for therapeutic purposes. There is a growing need for further research to investigate how different signaling pathways that regulate SASP, such as p38MAPK, mTOR, GATA4, TAK1, cGAS/cGAMP/STING, are interconnected and how SASP manifests aging-related pathologies.
Inhibition of SASP without disrupting stable growth arrest would reduce deleterious effects while maintaining tissue homeostasis and other physiological roles. However, targeting SASP for therapeutic purposes must be undertaken with great care, as it has both beneficial and deleterious roles due to the plethora of components. Identification of key SASP factors secreted by senescent cells in aged tissues and residual tumors in the post-treatment period could have potential as biomarkers for real-time medical surveillance.
This very intriguing topic has just been reported by the authors and is an established finding in the literature. Authors should deepen this aspect in the discussion.
The third step is to identify factors that are expected to be altered in the specific context. Transcriptome and proteome profiling of individual tissue cells, along with the development of sophisticated high-throughput methods and machine learning tools, will be critical to understanding the nature of senescent cells and may help identify potential therapeutic approaches. A new database, SeneQuest1, has been created to help identify genes associated with senescence.
This is a very interesting topic to address.
A recent interesting study by Martínez-Zamudio et al. (2020) revealed the links between enhancer chromatin, transcription factor recruitment and senescence competence. They demonstrated that a hierarchical network of transcription factors defines the transcriptional program of senescence and identified activator protein 1 (AP-1) as a master regulator driving the transcriptional program of senescent cells, thus revealing promising pathways with therapeutic implications for the modulation of senescence in vivo.
Finally, accumulating evidence has shown that both anti-senescence and pro-senescence therapies may be useful depending on the context. Pro-senescence therapies help to limit damage by containing proliferation and fibrosis during carcinogenesis and active tissue repair, while anti-senescence agents allow elimination of accumulated senescent cells to restore tissue function and potentially aid organ rejuvenation.
It has been found that cells that escape senescence after chemotherapy re-enter the cell cycle, are highly aggressive, resistant to chemotherapy, exhibit stem cell characteristics, and may contribute to cancer recurrence. Because different therapeutic modalities trigger senescence in tumors, it is important to decipher the mechanisms involved in senescence escape, as a more detailed understanding may enable the development of better therapies and help reduce off-target effects that contribute to unwanted toxicity.
Author Response
To Reviewer #3
Thank you very much for your criticism and many useful comments. Our point-by-point answers to your comments are as follows. The revised text is shown in blue in the revised manuscript.
The authors report in the abstract, "Cells undergoing irreversible cell cycle arrest that do not undergo cell division are believed to be significantly inactive in terms of energy metabolism due to unnecessary energy consumption for cell division.". They add that evidence has been obtained in their study that overturns this prevailing orthodoxy. Their statement is too simplistic and is not reflected in the more recent but established literature. Many papers in the literature report the complex process that accompanies cell cycle arrest induced by radiation exposure and in general when senescence processes occur.
Our answer: We agree with the reviewer’s comment, and the first sentence of abstract has been rewritten as
"Irreversible cell-cycle arrested cells not undergoing cell divisions have been thought to be meta-bolically less active because of unnecessary consumption of energy for cell division. On the other hand, they might be active to be involved in the tissue microenviroment through an inflammatory response. In this study, we examined mitochondria-dependent metabolism in human cells irreversibly arrested in response to ionizing radiation to confirm the possibility. "
Besides, in the introduction section, we have cited several recent literatures and gave overview of the current status at the first paragraph of Introduction.
Many articles and reviews report the different mechanistic pathways and mediators underlying the finite proliferation of normal somatic cells and how entry into senescence leading to stable cell cycle arrest is regulated.
The authors are urged to downplay this claim, which is not reflected in the current literature.
Our answer: We deeply appreciate the comments, which should be taken into consideration. In the revised manuscript, we have provided five additional references in the revised manuscript as follows, in order to expand our view of the mechanisms underlying irreversible cell cycle arrest.
- Ewald, J.A.; Desotelle, J.A.; Wilding, G.; Jarrard, D.F. Therapy-induced senescence in cancer. J Natl Cancer Inst 2010, 102, 1536-1546, doi:10.1093/jnci/djq364.
- Saleh, T.; Bloukh, S.; Carpenter, V.J.; Alwohoush, E.; Bakeer, J.; Darwish, S.; Azab, B.; Gewirtz, D.A. Therapy-Induced Se-nescence: An "Old" Friend Becomes the Enemy. Cancers (Basel) 2020, 12, doi:10.3390/cancers12040822.
- Blagosklonny, M.V. Cell senescence, rapamycin and hyperfunction theory of aging. Cell Cycle 2022, 21, 1456-1467, doi:10.1080/15384101.2022.2054636.
- Blagosklonny, M.V. Hallmarks of cancer and hallmarks of aging. Aging (Albany NY) 2022, 14, 4176-4187, doi:10.18632/aging.204082.
- Dimri, G.P.; Lee, X.; Basile, G.; Acosta, M.; Scott, G.; Roskelley, C.; Medrano, E.E.; Linskens, M.; Rubelj, I.; Pereira-Smith, O. A biomarker that identifies senescent human cells in culture and in aging skin in vivo. Proc Natl Acad Sci U S A 1995, 92, 9363-9367, doi:10.1073/pnas.92.20.9363.
Thus, the paper does not address a new and scientifically relevant topic and that said, the work loses much of the claimed scientific originality, but it may be still an interesting work because it proposes a possible marker, namely the different dependance of Delta membrane potential DΨ of dose exposure. This part of the study is certainly more interesting and deserves more in-depth study than the observational-only description.
The authors should stress this part and further investigate the possible implications of DΨ behavior as a function of time after irradiation and better justify the different behavior of the two lines that had instead shown close similarities for mean mitochondrial area and validation tests of irreversible cell-cycle arrest by EdU and SA-β-gal staining, after irradiation
Our answer: We thank you again for the valuable comments and evaluation. According to the reviewer’s suggestion, we further discussed DΨ dynamics by adding new sentences in the third paragraph of Discussion in the revised manuscript as follows.
“Although the high ΔΨm area was overshot for the primary cells (WI-38) 5 days after irradiation, the fraction of the high ΔΨm area retained for the immortalized BJ-5ta cells. Low fraction of high ΔΨm area was also reported for another immortalized human fibroblast cells, BJ-1 h-TERT (Kaminaga et al. 2020). These evidences strongly suggest that energy metabolism is upregulated in a delayed manner in primary cells, and this is not the case for immortalized cells. Conversely, the manipulation of immortalization might alter the energy metabolism relevant to stress responses.”
The validation tests, recommended by the reviewer, have already been performed. Thus their results are shown in the revised manuscript. Please see the section of “2.2 Validation of irreversible cell-cycle arrest by EdU and SA-β-gal staining” in Results.
Furthermore, although the circumvention of senescence and the acquisition of unlimited replicative potential is a key event necessary for malignant transformation, the underlying signaling pathways and the basis for the stability of growth arrest are poorly understood. Therefore, this paperi is welcome. Greater understanding is therefore essential if we are to prevent tissue dysfunction without increasing the risk of developing cancer.
The authors are invited to make a connection with the possible implications of their findings in terms of preventing carcinogenesis.
Our answer: We would like to thank the reviewer again for the thoughtful comments, which apparently show the path towards the future study. Our next experiments have been undertaken to connect the current observations with the future prevention of cancer.
There is also much room for further progress in understanding the mechanisms underlying transient and short-lived senescence that promotes tissue development, regeneration, and repair, as it is less well characterized than the deleterious effects of stable senescence. One of the major obstacles in the field of senescence is the lack of a single universal and robust biomarker that can identify senescent cells with high sensitivity and specificity and that can differentiate them from terminally differentiated, quiescent, and other nondividing cells.
Our answer: We completely agree with the comments, and we are always searching for the robust biomarkers for identifying senescent cells even in the tissue samples.
Growth arrest is a key feature that can be easily demonstrated in vitro using colony formation assays or BrdU/EdU incorporation assays that measure DNA synthesis. However, measurement of DNA synthesis is not entirely specific, as DNA repair may still be active. Measuring the expression levels of CDKIs p16INK4A and p21WAF1=CIP1 is critical for detecting cell cycle arrest but is not persistently expressed by senescent cells. The accumulation of high levels of p16INK4A is necessary to maintain the senescent state by preventing RB inactivation, which allows it to be used extensively as a senescence marker in most normal nontransformed cells and tissues. Because of the heterogeneous and dynamic nature of senescence, there is currently no single totally reliable biomarker. Recently, a three-step multimarker workflow has been proposed to accurately detect senescent cells.
The first stage includes the assessment of senescence activity associated with beta-galactosidase (SA-b-gal) and/or lipofuscin (GL-13 or SBB) accumulation. The authors reported experiments on the detection of SA-beta-gal).
The second phase examines frequently observed markers of senescent cells, including transcriptional signatures related to cell cycle arrest and SASP, such as increased expression of cyclin-dependent kinase inhibitors and a subset of SASP genes, along with decreased expression of proliferation markers such as cyclins, CCNA2 and CCNE2, and LMNB1. A deeper understanding of the underlying mechanisms that regulate senescence will provide promising translational opportunities to develop novel therapeutic approaches that minimize the harmful consequences of senescence. Targeting senescence using senolytics to selectively eliminate senescent cells or modulating SASP using small molecules or antibodies will not only help treat senescence-related diseases, but may help improve the healthspan of individuals.
A thorough understanding of the regulation of SASP is needed to harness it for therapeutic purposes. There is a growing need for further research to investigate how different signaling pathways that regulate SASP, such as p38MAPK, mTOR, GATA4, TAK1, cGAS/cGAMP/STING, are interconnected and how SASP manifests aging-related pathologies.
Inhibition of SASP without disrupting stable growth arrest would reduce deleterious effects while maintaining tissue homeostasis and other physiological roles. However, targeting SASP for therapeutic purposes must be undertaken with great care, as it has both beneficial and deleterious roles due to the plethora of components. Identification of key SASP factors secreted by senescent cells in aged tissues and residual tumors in the post-treatment period could have potential as biomarkers for real-time medical surveillance.
This very intriguing topic has just been reported by the authors and is an established finding in the literature. Authors should deepen this aspect in the discussion.
Our answer: We deeply appreciate the reviewer to provide such thoughtful and broad discussions. It must be a compass showing the research direction of not only us but also the relevant scientists. Then, we have included them in our discussion. Please see blue sentences in Discussion in the revised manuscript.
The third step is to identify factors that are expected to be altered in the specific context. Transcriptome and proteome profiling of individual tissue cells, along with the development of sophisticated high-throughput methods and machine learning tools, will be critical to understanding the nature of senescent cells and may help identify potential therapeutic approaches. A new database, SeneQuest1, has been created to help identify genes associated with senescence.
This is a very interesting topic to address.
Our answer: Thanks again for providing updated information, which are really those considered as the future study. We are hoping that such novel techniques enable to connect our current observation with molecular-based understandings of the mitochondrial functional change.
A recent interesting study by Martínez-Zamudio et al. (2020) revealed the links between enhancer chromatin, transcription factor recruitment and senescence competence. They demonstrated that a hierarchical network of transcription factors defines the transcriptional program of senescence and identified activator protein 1 (AP-1) as a master regulator driving the transcriptional program of senescent cells, thus revealing promising pathways with therapeutic implications for the modulation of senescence in vivo.
Finally, accumulating evidence has shown that both anti-senescence and pro-senescence therapies may be useful depending on the context. Pro-senescence therapies help to limit damage by containing proliferation and fibrosis during carcinogenesis and active tissue repair, while anti-senescence agents allow elimination of accumulated senescent cells to restore tissue function and potentially aid organ rejuvenation.
It has been found that cells that escape senescence after chemotherapy re-enter the cell cycle, are highly aggressive, resistant to chemotherapy, exhibit stem cell characteristics, and may contribute to cancer recurrence. Because different therapeutic modalities trigger senescence in tumors, it is important to decipher the mechanisms involved in senescence escape, as a more detailed understanding may enable the development of better therapies and help reduce off-target effects that contribute to unwanted toxicity.
Our answer: Once again, we deeply appreciate the reviewer's thoughtful and broader perspective discussion. We do hope to be able to integrate these issues into our future experiments.
Reviewer 4 Report (New Reviewer)
The authors, in this work, show the radiation effect on fibroblasts evaluating cellular cycle and mitochondrial state. The authors declare in the abstract that “Irreversible cell-cycle arrested cells not undergoing cell divisions have been thought to be considerably inactive in terms of energy metabolism […]”; therefore, the authors, write that the aim of their work is to overturn this assumption. They declare to demonstrate that, after radiation, mitochondria keep metabolic activity, despite cycle cell is stopped.
General comments
The work shows serious flaws, starting from the aim that is not clear. The results present deep weaknesses that do not allow to get solid scientific conclusions, neither any speculations. A poor discussion, based on the description of some results mentioning figures, does not clarify data; moreover, the absence of conclusions, does not allow the reader to comprehend the scientific meaning of the work that, overall, is not acceptable for the publication.
Starting to read, the aim of the work is not clear. The authors present the work introducing the effect of radiotherapy on cycle cell arrest that, on survival cancer cells, could be a prominent factor on development of new carcinogenesis. However, the analysis are conducted on fibroblasts. For this reason, it must be clarified whether the authors wanted to better investigate the off-target effects of ionizing radiation on stromal cells. This aspect is fundamental considering that all the reported results cannot be translate to cancer cells that, have a completely different metabolism, compared to healthy cells.
Specific comments
· Lines 45-49: the authors should introduce concepts about mitochondrial metabolism in cancer rather than in other pathologies.
· Lines 47-55: the description on cellular volume and ATP production in cell cycle arrest is confusing. These introducing concepts are essential for the purpose of the work and the reader can’t understand the starting points.
· Figure 1: In the first one, standard deviation is not indicated for 20 Gy. In both figures, statistical analysis is not performed. The meaning of this experiment is not clarified. What do the authors want to demonstrate by reporting that cells treated with 20 Gy reduce proliferation? Moreover, they describe this figure in the results, writing: “WI-38 and BJ-5ta cells did not show such catastrophic cell death and most of the cells (90%) were alive without cell-cycle progression”. In which experiment is this demonstrated?
Images in figure 2 have a bad resolution, scale bars are not indicated, cells are deformed as if the box had been resized.
· Statistical analysis and scale bars are absent in following figures.
The absence of scale bars excludes the possibility to demonstrate the results about cellular hypertrophy since it appears that the authors enlarged the images in Figure 4A.
· In the experiments, controls (0 Gy or sham irradiated) at day 2, 5 and 10 are absent.
· Interpretation and discussion of results are not clear. After reading the entire article, I cannot understand the signification that authors want to give to radiation effects in relation with cycle cell arrest and mitochondrial activity. In the discussion, lines 228-234, is reported a summary about the general meaning of the article but the data reported in results are not solid enough to support these statements.
· English language requires a major revision.
Author Response
To Reviewer #4
The authors do appreciate the reviewer's comments and his/her critical reading of the manuscript. We carefully revised the manuscript to describe our findings with clarity with clear conclusions and sufficient discussions. We substantially rewrote the original sentences, or added new sentences, as well as cited additional references. These modifications are shown in purple in the revised manuscript. We also changed the figures to much clear ones according to the reviewer’s comments. Our point-by-point answers to your comments are as follows.
General comments
The work shows serious flaws, starting from the aim that is not clear. The results present deep weaknesses that do not allow to get solid scientific conclusions, neither any speculations. A poor discussion, based on the description of some results mentioning figures, does not clarify data; moreover, the absence of conclusions, does not allow the reader to comprehend the scientific meaning of the work that, overall, is not acceptable for the publication.
Starting to read, the aim of the work is not clear. The authors present the work introducing the effect of radiotherapy on cycle cell arrest that, on survival cancer cells, could be a prominent factor on development of new carcinogenesis. However, the analysis are conducted on fibroblasts. For this reason, it must be clarified whether the authors wanted to better investigate the off-target effects of ionizing radiation on stromal cells. This aspect is fundamental considering that all the reported results cannot be translate to cancer cells that, have a completely different metabolism, compared to healthy cells.
Our answer: We deeply appreciate the comment, as it is critical point to be considered. We rewrote the manuscript, particularly the paragraph in Introduction by citing new references, to emphasize that the study has used fibroblasts not cancer cells. We intended to provide description on the significance of senescent fibroblasts in tumor microenvironments which could receive undesired exposure of radiation during radiotherapy. It needs a substantial space to list each modified sentence here. So please see purple sentences in the manuscript, particularly in Introduction.
Specific comments
- Lines 45-49: the authors should introduce concepts about mitochondrial metabolism in cancer rather than in other pathologies.
Thank you very much for the comment. We did this accordingly.
- Lines 47-55: the description on cellular volume and ATP production in cell cycle arrest is confusing. These introducing concepts are essential for the purpose of the work and the reader can’t understand the starting points.
Our answer: Thank you very much for the comment again. We rewrote the sentence accordingly.
- Figure 1: In the first one, standard deviation is not indicated for 20 Gy. In both figures, statistical analysis is not performed. The meaning of this experiment is not clarified. What do the authors want to demonstrate by reporting that cells treated with 20 Gy reduce proliferation? Moreover, they describe this figure in the results, writing: “WI-38 and BJ-5ta cells did not show such catastrophic cell death and most of the cells (90%) were alive without cell-cycle progression”. In which experiment is this demonstrated?
Our answer: Standard deviations were added in the graphs in Figure 1 as shown by error bars. A dose of 20 Gy is quite high and sometimes such high dose cause catastrophic cell death as determined by destruction of cellular shapes.
We did not recognize such disruptive effect in the wide-field microscopic images of both cell lines as shown in the microscopic images in Figure 1. The number of DAPI-stained cells in the images were counted and it kept almost the same number of it counted before 20 Gy X-ray exposure. This is the evidence clearly suggested that the high dose exposure did not significantly cause the catastrophic cell death. To clearly describe this, we complemented some phrases in the section “2.1 Cell proliferation against X-ray dose” as shown by purple.
Images in figure 2 have a bad resolution, scale bars are not indicated, cells are deformed as if the box had been resized.
Our answer: We replaced the images with much clear and high-resolution ones in the revised manuscript. Scale bars were now indicated to all figures in the manuscript to show an enlarged and flattened morphology, which are phenotypes presented by irreversible cell-cycle-arrested cells. Also, all figures in the manuscript were re-checked and some figures were replaced with clear figures according to reviewer’s comment.
- Statistical analysis and scale bars are absent in following figures.
The absence of scale bars excludes the possibility to demonstrate the results about cellular hypertrophy since it appears that the authors enlarged the images in Figure 4A.
Our answer: We performed statistical analysis for the figures and added scale bars to all figures accordingly.
- In the experiments, controls (0 Gy or sham irradiated) at day 2, 5 and 10 are absent.
Our answer: Prior to the experiments, we confirmed that the fraction in the control cells did not change over 10 days. Besides, culturing non-irradiated cells for 9 days without subculturing lead cells to become confluent and this would give us the results of reversible-arrested cells as the control, which do not match to our aim of the study.
- Interpretation and discussion of results are not clear. After reading the entire article, I cannot understand the signification that authors want to give to radiation effects in relation with cycle cell arrest and mitochondrial activity.
Our answer: We completely agree with the reviewer’s comment. To make clear discussion on the cell cycle arrest and mitochondrial activity, we added sentences to the third paragraph in Discussion as follows. Some additional sentences are shown in blue (for another reviewer who also gave us a similar comments).
“The normalized fraction of the high ΔΨm area of WI-38 and BJ-5ta cells visual-ized by JC-1 staining reached its lowest level at 2 days after irradiation (Figure 5B). This observation suggests that mitochondrial activity was temporarily decreased due to the degradation of damaged mitochondria (as indicated above) as reported previ-ousely [14], and then the high ΔΨm area was re-established within a few days. Alt-hough the high ΔΨm area was overshot for the primary cells (WI-38) 5 days after irra-diation, the fraction of the high ΔΨm area retained for the immortalized BJ-5ta cells. Low fraction of high ΔΨm area was also reported for another immortalized human fi-broblast cells, BJ-1 h-TERT [29](Kaminaga et al. 2020). These evidences strongly sug-gest that energy metabolism is upregulated in a delayed manner in primary cells, and this is not the case for immortalized cells. Conversely, the manipulation of immortali-zation might alter the energy metabolism relevant to stress responses.”
(Please note that these sentences might be edited by the Journal English Correction Service in the manuscript body.)
In the discussion, lines 228-234, is reported a summary about the general meaning of the article but the data reported in results are not solid enough to support these statements.
Our answer: We modified the sentences with additional references as follows.
“Since premature senescence in tumor microenvironment has extensively been dis-cussed in terms of the treatment efficacy, manifestation of adverse effects, as well as the QOL of the patients [37-42], a comprehensive understanding of the radiation bio-logical significance of irreversible cell-cycle arrest after radiotherapy is expected.”
(Please note that this sentence might be edited by the Journal English Correction Service in the manuscript body.)
- English language requires a major revision.
We used a paid Journal English Correction Service.
Round 2
Reviewer 3 Report (New Reviewer)
The authors partially responded to the querys by arguing with modification of the initial text.Some of the comments were left unanswered because responding would have involved a lot of additional work, and the hope is that the authors will take this into account for possible future work. As I already said in the first reviewing although not very original may be useful to the scientific community for some of the results reported. The authors have agreed to tone down their statements. The paper in this form may be accepted for publication in IJMS
Author Response
Thank you very much for your thoughtful and encouraging words. We will continue the current study and solve the unanswered questions in the future work.
Reviewer 4 Report (New Reviewer)
General comments
The authors have improved the introduction and the aim of the work, which is now more clear and understandable. In the introduction, the authors describe the importance of the study considering the correlation between cancer and stromal cells. In addition to pancreatic cancer, another tumor characterized by a particularly complex microenvironment is glioblastoma. Therefore, this study could also open important scenarios for the treatment of this tumor, which is also targeted by ionizing radiation with an aggressive multimodal treatment. The authors could mention this.
In summary, the authors have solved most of the flaws and the quality of the work has slightly improved. Other critical issues of the work are still not resolved that the authors could fix with minor revisions.
Specific comments
2. Lines 45-49: the authors should introduce concepts about mitochondrial metabolism in cancer rather than in other pathologies.
OK, the authors solved this concern.
3. Lines 47-55: the description on cellular volume and ATP production in cell cycle arrest is confusing. These introducing concepts are essential for the purpose of the work and the reader can’t understand the starting points.
OK, the authors solved this concern.
4. Figure 1: In the first one, standard deviation is not indicated for 20 Gy. In both figures, statistical analysis is not performed. The meaning of this experiment is not clarified. What do the authors want to demonstrate by reporting that cells treated with 20 Gy reduce proliferation? Moreover, they describe this figure in the results, writing: “WI-38 and BJ-5ta cells did not show such catastrophic cell death and most of the cells (90%) were alive without cell-cycle progression”. In which experiment is this demonstrated?
The authors wrote that cell proliferation was significantly suppressed in 20 Gy X-irradiated WI-84 and BJ-5ta cells but both cell lines did not show such catastrophic cell death and 90% were present in culture without cell-cycle progression. These seem to be contradictory results. 90% after irradiation is very high to demonstrate such a difference reported in graphs compared to 0 Gy. The authors could insert simple light field photos since DAPI quality is not so good, even if supplementary comparing 0 and 20 Gy over time. It would have been more appropriate to perform a low cell density assay analysis such as clonogenic assay, which is a gold standard for ionizing radiation effects, especially for long incubation times.
5. Images in figure 2 have a bad resolution, scale bars are not indicated, cells are deformed as if the box had been resized.
Images quality and resolution is still poor.
6. Statistical analysis and scale bars are absent in following figures. The absence of scale bars excludes the possibility to demonstrate the results about cellular hypertrophy since it appears that the authors enlarged the images in Figure 4A.
OK, the authors solved this concern. However, I suggest using alternative tests to t-Student in the case of comparing multiple samples such as ANOVA or another test if the data have no Gaussian distribution. In this regard, a normality test should be conducted.
7. In the experiments, controls (0 Gy or sham irradiated) at day 2, 5 and 10 are absent.
This problem can be solved with a clonogenic assay as discussed above. By culturing the cells at a low density, the critical issues associated with confluence would be solved.
8. Interpretation and discussion of results are not clear. After reading the entire article, I cannot understand the signification that authors want to give to radiation effects in relation with cycle cell arrest and mitochondrial activity.
OK, the authors solved this concern. However, it would be appropriate to emphasize that future studies should clarify metabolic aspects. Therefore, in further studies related with this approach It would be appropriate to perform oxygen consumption rate extracellular acidification rate of live cells investigating key cellular functions such as mitochondrial respiration and glycolysis
9. In the discussion, lines 228-234, is reported a summary about the general meaning of the article but the data reported in results are not solid enough to support these statements.
OK, the authors solved this concern.
10. English language requires a major revision.
OK, the authors solved this concern.
Author Response
Please see attached the file.

This manuscript is a resubmission of an earlier submission. The following is a list of the peer review reports and author responses from that submission.
Round 1
Reviewer 1 Report
This paper describes the change in mitochondrial content and membrane potential in X irradiated cells.
The results described are interesting as they report new data on mitochondria dysfunction over a rather long time period. However I find that the discussion is really too short. It would be needed to discuss the results in the general framework of other papers dealing with similar issues.
In the introduction, the dose of 20 Gy which is very high, should be justified. How is this dose relevant to mimic senescence ?
Detailed remarks:
Line 54 : which leads "to" irreversible effect
line 72 change "that whether" by "if"
Figure 2A is too dark
Line 139 : in terms of their function as revealed...
Line 156 : growth arrested cells "are" known
Material and methods
4.1 : edit correctly CO2 and figures
Line 197 : "The" excitation and emission
Line 202 : Phase contrast images "were" taken along
Line 225 : Cells in "their" observed microscope
Line 257 : split in 2 sentences
Line 268 : we report mitochondrial effects ...
Line 269 : the evidences indicate that "despite" the high dose
Author Response
Thank you very much for your understanding of our study and your valuable suggestions, particularly for the reinforce strengthening of the Discussion section. We addressed this in the revised manuscript in terms of the “effect of radiation on mitochondria” and “cellular senescence” to improve the Discussion.
Our answers to your comments are listed in the attached file. The changes made in response to your comments are shown in purple in the revised manuscript.

Reviewer 2 Report
The manuscript entitled " Change of Mitochondrial Content and Its Membrane Potential in the X-Irradiated Cells Undergoing Senescence-Like Growth Arrest" described the mechanism underlying senescence-like growth arrest after X-irradiation focusing on the mitochondrial content and function. I carefully reviewed the manuscript, but I regret to say that there are some logical weak points in it; for example, lack of statistical significance made it impossible to lead conclusion. Moreover, the conclusion of the manuscript was not scientifically and sufficiently supported by their data. I, therefore, could not be positive on this manuscript for publication in International Journal of Molecular Sciences.
Major points:
- Firstly, all experiments were conducted using immortalized foreskin fibroblast cells only. It would be critical to use multiple kinds of cell lines in each experiment to demonstrate generality of the conclusion the authors declared.
- The authors pointed the re-activation or re-synthesis of mitochondria comparatively long time after X-irradiation with emphasis as a novelty of this study. I agree this opinion, but the data of Figs. 3 &4 which should confirm this finding, are very weak without statistical significance. Hence, I’m not sure that the authors’ claim is true.
- As the evidence for occurrence of senescence, only SA-β-gal staining is not sufficient. It is necessary to show the additional supportive data such as the induction of SASP factors.
- To show that increased mitochondrial mass is important for maintaining cell function in senescent cells, authors should perform the additional analysis as follows,
・ Check the state of mitochondria during senescence induction due to stress other than radiation
・ Examine how CCCP affects the induction of cell senescence when mitochondrial function is inhibited by losing membrane potential.
・ Test the cells with limited mitochondrial function and amount, such as ρ0 cells
- In addition to the mitochondrial membrane potential, ATP production in cells or cellular respiration is also the reliable indicator for mitochondrial function.
- As authors mentioned, 20 Gy is obviously high dose. Since the dose dependency is also important information, the authors should examine the smaller doses such as 5 and 10 Gy.
Author Response
Thank you very much for your careful review and many valuable comments. We understand your criticism in terms of insufficient statistical analysis of the data. We have replied to your comments and questions in a point-by-point manner as listed in the attached file, and have substantially revised our manuscript. The changes made in response to your comments are shown in red in the revised manuscript.

Reviewer 3 Report
Hirose et al. describe their work in the manuscript titled, “Change of mitochondrial content and its membrane potential in the X-irradiated cells undergoing senescence-like growth arrest.” In the beginning, the authors nicely validate senescence-like growth arrest with the help of SA-β-gal and Edu staining. Later, they investigate the mean mitochondrial area after irradiation and the mean proportion of the high mitochondrial membrane potential area during senescence. The work is exciting and certainly has potential but needs a lot of additional experiments to reach there. My main concerns outlined below,
- First of all, authors have used transformed cell line, thus not a good model to study cellular senescence. To study cellular senescence, it is always best to use primary cells and not transformed cells.
- Senescence is a heterogeneous phenotype, which means that not all cells are undergoing senescence and show the same phenotype. How do authors know the phenotype they observe in all cells and not in specific cells? If only particular cells, what is the percentage of those cells?
- Authors use only two methods, SA Beta gal and EdU incorporation, for characterizing senescence phenotype. How do they know that these cells are senescent other than beta gal labeling? SABeta gals are not the only marker for cellular senescence. They should show some qPCR data for senescence genes to show that these cells are senescent.
- Authors are using only one cell type for all their experiments. Would it be informative to test these identical phenotypes in other cell types such as lung fibroblast and primary cells?
Author Response
Thank you very much for your deep understanding our study. Our reply to the reviewer are listed in the attached file. We revised the manuscript according to your suggestions as shown in green text in the revised manuscript.

Round 2
Reviewer 2 Report
The author responded to my suggestions satisfactory and now the revised manuscript sounds more impressive. Therefore, I recommend it for publication.
Reviewer 3 Report
The authors have not clearly addressed any of the comments. ATCC gives primary cells which do not need any ethics committee as it is commercially available. I certainly do not appreciate working on the transformed cells for senescence phenotype as it does not provide any adequate evidence that it can happen in normal cells. I do not see the physiological relevance of the data and thus feel it inappropriate to be published without sufficient evidence.